# Efficient and thermally stable organic solar cells based on small molecule donor and polymer acceptor

Zijian Zhang[1,2], Junhui Miao[1,3], Zicheng Ding[1], Bin Kan[4], Baojun Lin[5], Xiangjian Wan[4], Wei Ma[5], Yongsheng Chen [iD] [4], Xiaojing Long[1], Chuandong Dou[1], Jidong Zhang[1], Jun Liu[1] & Lixiang Wang[1]

Efficient organic solar cells (OSCs) often use combination of polymer donor and small molecule acceptor. Herein we demonstrate efficient and thermally stable OSCs with combination of small molecule donor and polymer acceptor, which is expected to expand the research field of OSCs. Typical small molecule donors show strong intermolecular interactions and high crystallinity, and consequently do not match polymer acceptors because of large-size phase separation. We develop a small molecule donor with suppressed π-π stacking between molecular backbones by introducing large steric hindrance. As the result, the OSC exhibits small-size phase separation in the active layer and shows a power conversion efficiency of 8.0%. Moreover, this OSC exhibits much improved thermal stability, i.e. maintaining 89% of its initial efficiency after thermal annealing the active layer at 180 °C for 7 days. These results indicate a different kind of efficient and stable OSCs.

---

[1] State Key Laboratory of Polymer Physics and Chemistry, Changchun Institute of Applied Chemistry, Chinese Academy of Sciences, 130022 Changchun, P. R. China. [2] University of Chinese Academy of Sciences, No.19A Yuquan Road, 100049 Beijing, P. R. China. [3] University of Science and Technology of China, 230026 Hefei, P. R. China. [4] Key Laboratory for Functional Polymer Materials and Centre for Nanoscale Science and Technology, Institute of Polymer Chemistry, College of Chemistry, Nankai University, 300071 Tianjin, China. [5] State Key Laboratory for Mechanical Behavior of Materials, Xi'an Jiaotong University, 710049 Xi'an, P. R. China. Correspondence and requests for materials should be addressed to Z.D. (email: dean132@ciac.ac.cn) or to W.M. (email: msewma@xjtu.edu.cn) or to Y.C. (email: yschen99@nankai.edu.cn) or to J.L. (email: liujun@ciac.ac.cn)

Organic solar cells (OSCs) have received widespread attentions because of their potential as cheap and flexible photovoltaic technology[1–3]. The power conversion efficiency (PCE) of single-junction OSCs has increased to 16%[4,5], which is acceptable for the commercial applications. However, the stability issue remains to be a great obstacle for the practical applications of OSCs[6–8]. During the long-term outdoor operation of OSCs, the repeated thermal annealing/cooling effect is inevitable, which may disturb the active layer morphology and decrease the photovoltaic efficiency. Therefore, the morphology stability of the active layer under thermal stress is a key topic for the practical application of OSCs.

The active layer of OSCs consists of a blend of p-type organic semiconductor as electron donor and n-type organic semiconductor as electron acceptor. Either small molecules or polymers can be used as electron donor or electron acceptor in the active layer. At present, the mainstream of OSCs use the combination of polymer donor and non-fullerene small molecule acceptor ($P_D$/$M_A$-type)[9–16], which have greatly boosted the performance of OSCs. On the contrary, OSCs based on the combination of small molecule donor and polymer acceptor ($M_D$/$P_A$-type) suffer from the low device efficiency (PCE < 5%)[16–22]. In this manuscript, we report efficient $M_D$/$P_A$-type OSCs with PCE over 8%. Most importantly, we demonstrate the superior morphology stability of the active layer based on $M_D$/$P_A$ blend under thermal stress.

One of the limitations with the $M_D$/$P_A$-type OSCs is the poor active layer morphology. Typical small molecule donors do not match polymer acceptors because of their strong intermolecular interactions and high crystallinity[23–26]. When they are blended with polymer acceptors, the blends often exhibit large-size phase separation, which limits the exciton diffusion/dissociation and results in low device efficiency in OSCs[17,27]. To circumvent this problem, we develop a small molecule donor bearing out-of-plane bulky substituents with suppressed π-π stacking between the molecular backbones. Another obstacle for $M_D$/$P_A$-type OSCs is the lack of polymer acceptors. The only widely used polymer acceptor is poly[(N,N′-bis(2-octyldodecyl)-naphthalene-1,4,5,8-bis(dicarboximide)-2,6-diyl)-alt-5,5′-(2,2′-bithiophene)] (N2200). This polymer acceptor possesses high electron mobility, but suffers from the low absorption coefficient and small exciton diffusion length[28,29]. To further improve the performance of $M_D$/$P_A$-type OSCs, different polymer acceptors should be employed. Recently, we have developed a class of polymer acceptors using boron-nitrogen coordination bond (B ← N) with high electron mobility and tunable absorption properties[30–33]. This enable us to select suitable polymer acceptors containing B ← N for $M_D$/$P_A$-type OSCs.

To fabricate $M_D$/$P_A$-type OSCs, we develop (5Z,5′Z)-5,5′-((5″,5″″′-(4,8-bis(9-(2-ethylhexyl)-9H-carbazol-3-yl)benzo[1,2-b:4,5-b′]dithiophene-2,6-diyl)bis(3,3″-dioctyl-[2,2′:5′,2″-terthiophene]-5″,5-diyl))bis(methanylylidene))bis(3-ethyl-2-thioxothiazolidin-4-one) (DR3TBDTC) as the small molecule donor and select poly[5,10-bis(2-decyltetradecyl)-4,4,9,9-tetrafluoro-7-methyl-2-(5-(2,3,5,6-tetrafluoro-4-(5-methylthiophen-2-yl)phenyl)thiophen-2-yl)-4,5,9,10-tetrahydro-3a,5,8a,10-tetraaza-4,9-diborapyrene-3a,8a-diium-5,11-diuide] (PBN-11) as the polymer acceptor[34]. The $M_D$/$P_A$-type OSC based on DR3TBDTC and PBN-11 exhibits small-size phase separation in the active layer and shows a PCE of 8.0%. Moreover, the $M_D$/$P_A$-type OSC device can maintain 89% of its initial PCE after thermal annealing the active layer at 180 °C for 7 days. This thermal stability is much superior to that of typical $P_D$/$M_A$-type OSCs. The demonstration of high-performance $M_D$/$P_A$-type OSCs is expected to greatly expand the research field of OSCs.

## Results

**Materials.** The chemical structures of DR3TBDTC and PBN-11 are shown Fig. 1. PBN-11 ($M_n$ and Đ are 40.8 kDa and 2.2) was synthesized following our previous method[34]. The synthetic route of DR3TBDTC is shown in Supplementary Fig. 1 and the

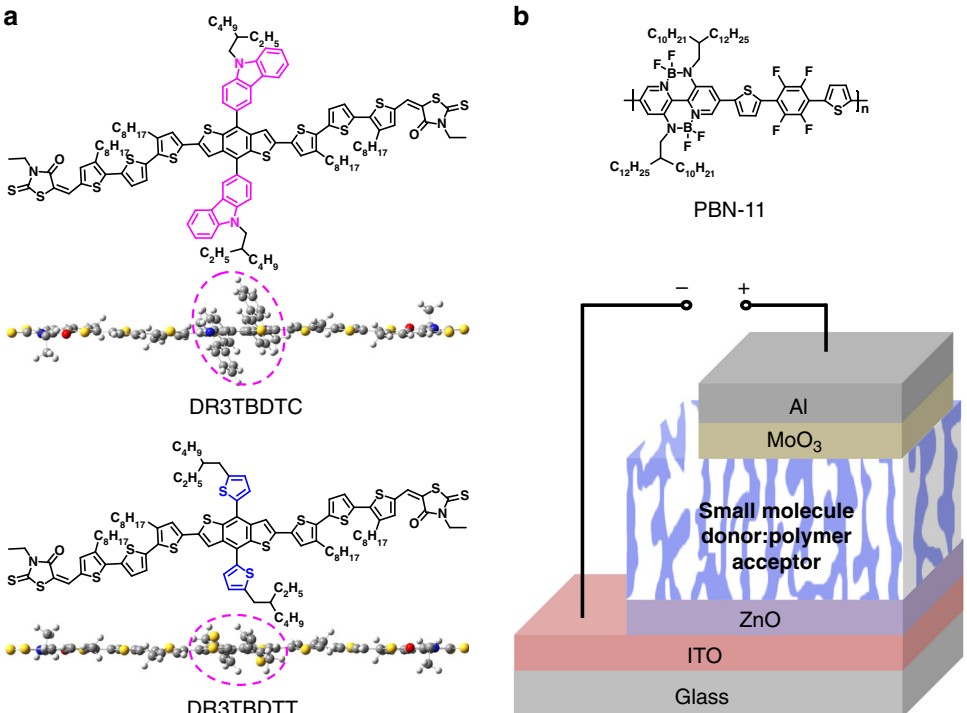

**Fig. 1** Photovoltaic materials and device structure of the OSCs. **a** Chemical structures of small molecule donors DR3TBDTC and DR3TBDTT, and the corresponding front view of the optimized geometries using DFT calculation (B3LYP/6-31 G*). **b** Chemical structure of polymer acceptor PBN-11 and the inverted device structure

synthesis procedures are provided in the Supplementary Methods. The chemical structure of DR3TBDTC is verified by [1]H NMR, [13]C NMR, mass spectrum and elemental analysis (Supplementary Figs. 2–4). The decomposition temperature of DR3TBDTC is as high as 406 °C, indicating the good thermal stability (Supplementary Fig. 5). To elucidate the effect of the bulky out-of-plane carbazolyl substituents in DR3TBDTC, we select a commercially available control compound, (5Z,5′Z)-5,5′-((5″,5″″-(4,8-bis(5-(2-ethylhexyl)thiophen-2-yl)benzo[1,2-b:4,5-b′]dithiophene-2,6-diyl)bis(3,3″-dioctyl-[2,2′:5′,2″-terthiophene]-5″,5-diyl))bis(methanylylidene))bis(3-ethyl-2-thioxothiazolidin-4-one) (DR3TBDTT)[35] (Fig. 1a). DR3TBDTT and DR3TBDTC have the similar chemical structure except the pendent carbazolyl/thienyl substituents. When used as the small molecule donor, DR3TBDTT works well with fullerene acceptors but does not match polymer acceptors in OSCs[27,36].

**Molecular geometry and physical properties**. We studied the molecular geometry of DR3TBDTC and DR3TBDTT by density functional theory (DFT) calculations[37]. As shown in Fig. 1a, both DR3TBDTC and DR3TBDTT exhibit nearly planar conjugated backbones. The dihedral angle between the carbazolyl/thienyl substituents and the benzodithiophene core in DR3TBDTC/DR3TBDTT is 58º/53º (Fig. 1a, Supplementary Figs. 6 and 7). Therefore, the carbazolyl/thienyl substituents act as steric hindrance to prevent close stacking of the planar conjugated backbones. As the carbazolyl substituents in DR3TBDTC are bulkier than the thienyl substituents in DR3TBDTT, DR3TBDTC is expected to show suppressed intermolecular interactions between the conjugated backbones. This speculation is verified by the absorption spectra and two-dimensional grazing incidence wide-angle X-ray scattering (2D-GIWAXS) results. Though the two compounds show similar absorption spectra in dilute chlorobenzene (CB) solution, DR3TBDTC shows less redshifted absorption spectrum and weaker long-wavelength vibronic shoulder than DR3TBDTT in neat and blend films (Fig. 2a, Supplementary Figs. 8–9). These results suggest that DR3TBDTC exhibits weaker π-π stacking between the molecular backbones in solid phase than that of DR3TBDTT[35]. Supplementary Fig. 10 shows the 2D-GIWAXS patterns of the thin films of the two compounds. Compared with DR3TBDTT film, DR3TBDTC film shows relatively weaker (200), (300), and (010) reflections, but exhibits multiple reflections in the $q_{xy}$ direction between 0.9 and 1.5 Å$^{-1}$. According to their (010) reflection peaks, DR3TBDTC has the larger π–π stacking distance ($d_{\pi-\pi}$ is 3.88 Å) than that of DR3TBDTT ($d_{\pi-\pi}$ is 3.64 Å) (Supplementary Table 1), which indicates the weaker π-π stacking of DR3TBDTC molecules in film. Compared with DR3TBDTT, the weaker π-π interactions of

DR3TBDTC are attributed to the bulkier out-of-plane carbazolyl substituents.

Despite of the suppressed π–π stacking, DR3TBDTC still shows high crystallinity and good charge transporting property. According to the differential scanning calorimetry (DSC) measurement (Fig. 2b and Table 1), DR3TBDTC exhibits the crystallization temperature ($T_c$) and crystallization enthalpy change ($\Delta H_c$) of 223.3 °C and 52.2 J g$^{-1}$, which is higher than those of DR3TBDTT ($T_c$ is 208.6 °C, $\Delta H_c$ is 32.0 J g$^{-1}$). These results indicate that DR3TBDTC exhibits higher crystallinity than DR3TBDTT. As estimated with the space-charge-limited current (SCLC) method, the hole mobilities ($\mu_h$) of DR3TBDTC and DR3TBDTT are 2.15 × 10$^{-4}$ and 3.37 × 10$^{-4}$ cm$^2$ V$^{-1}$ s$^{-1}$, respectively (Table 1 and Supplementary Fig. 11). The comparable hole mobilities indicate that the suppressed π–π stacking does not obviously deteriorate the hole-transporting property of DR3TBDTC. Moreover, the carbazolyl substituents do not show obvious effect on the molecular energy levels. The lowest unoccupied molecular orbital (LUMO)/highest occupied molecular orbital (HOMO) energy levels of DR3TBDTC (−3.09/−5.18 eV) are slightly higher than those of DR3TBDTT (−3.19/−5.21 eV) (Supplementary Fig. 12). The energy level alignment implies that they both can work as electron donors to match the polymer acceptor PBN-11[38,39].

**Photovoltaic properties**. The M$_D$/P$_A$-type OSCs were fabricated with an inverted device structure of ITO/ZnO/DR3TBDTC or DR3TBDTT:PBN-11/MoO$_3$/Al (Fig. 1b). The device optimization processes are shown in Supplementary Figs. 13–16 and Supplementary Tables 2–5. Fig. 3a displays the current density–voltage (J–V) plots for the OSC devices from the two blends with optimal conditions, and Table 2 shows the corresponding photovoltaic parameters. The open-circuit voltage ($V_{OC}$), short-circuit current density ($J_{SC}$) and fill factor (FF) for the control device based on DR3TBDTT:PBN-11 blend are 1.15 V, 6.21 mA cm$^{-2}$ and 42.9%, which yield a low PCE of 3.06%. This PCE is typical for M$_D$/P$_A$-type OSCs[18,21]. In comparison, the DR3TBDTC:PBN-11 based device shows a $V_{OC}$ of 1.11 V, a $J_{SC}$ of 11.18 mA cm$^{-2}$ and an FF of 64.6%, corresponding to a PCE of 8.01%. The J–V plot from the backward scan shows nearly identical PCE (see Supplementary Fig. 17 and Supplementary Table 6). To the best of our knowledge, this PCE is the highest value reported to date for M$_D$/P$_A$-type OSCs[19,40]. The much enhanced PCE of DR3TBDTC-based device is due to the greatly increased $J_{SC}$ and FF. In addition, the employment of larger side chains on the thienyl substituents cannot effectively improve the OSC performance (see Supplementary Fig. 18 and Supplementary Table 7), which suggest that the carbazolyl substituents play a crucial role in DR3TBDTC. Fig. 3b shows the external quantum efficiency

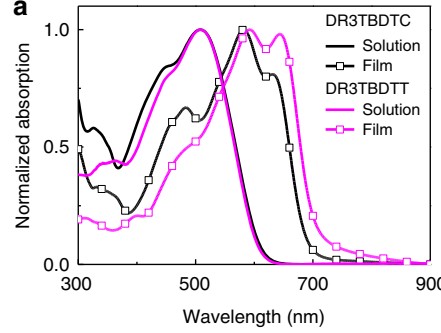
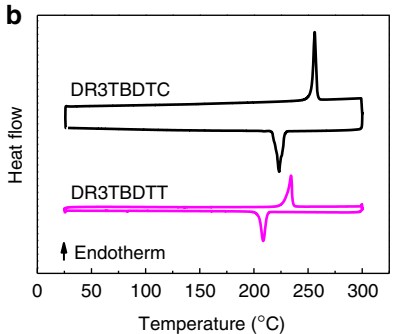

**Fig. 2** UV–vis absorption and DSC thermogram of small molecule donors. **a** Normalized absorption spectra of DR3TBDTC and DR3TBDTT in CB solution (1 × 10$^{-5}$ M) and in thin film (spin-coated from CB solution). **b** DSC second heating and cooling cycles of DR3TBDTC and DR3TBDTT in nitrogen atmosphere with a scan rate of 10 °C min$^{-1}$

**Table 1 The optical, electrochemical, thermal properties, $\pi$-$\pi$ stacking distance and hole mobility of DR3TBDTC and DR3TBDTT**

| Donors | $\lambda_{max}$[a] (nm) | $\lambda_{max}$[b] (nm) | $\varepsilon_{max}$[b] (cm$^{-1}$) | $E_g^{opt\,b}$ (eV) | $E_{HOMO}$[c] (eV) | $E_{LUMO}$[c] (eV) | $T_c$ (°C) | $\Delta H_c$ (J g$^{-1}$) | $d_{\pi\text{-}\pi}$ (Å) | $\mu_h$ (10$^{-4}$ cm$^2$ V$^{-1}$ s$^{-1}$) |
|---|---|---|---|---|---|---|---|---|---|---|
| DR3TBDTC | 508 | 582 | $1.06 \times 10^5$ | 1.77 | −5.18 | −3.09 | 223.3 | 52.2 | 3.88 | 2.15 (1.95 ± 0.15) |
| DR3TBDTT | 508 | 592 | $1.05 \times 10^5$ | 1.75 | −5.21 | −3.19 | 208.6 | 32.0 | 3.64 | 3.37 (3.04 ± 0.20) |

The hole mobility data in parentheses are the statistical average and error bars of standard deviation calculated from 16 individual devices and data outside of parentheses are the best results
[a]CB solution
[b]Thin films spin-coated from CB solution
[c]Cyclic voltammetry carried out on the as-cast thin films and the energy levels estimated by the equation of $E_{HOMO/LUMO} = -(4.80 + E_{onset}^{ox}/E_{onset}^{red})$ eV

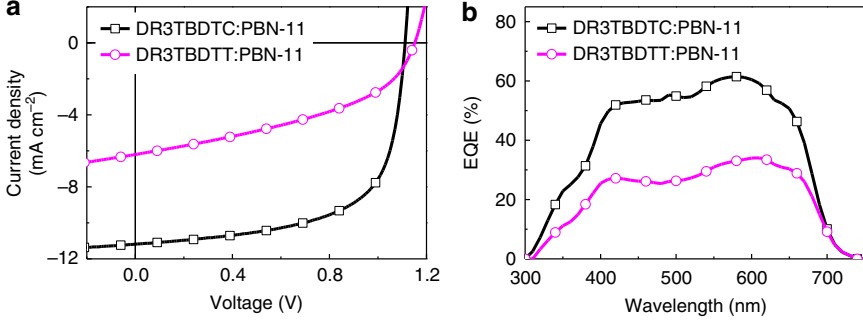

**Fig. 3** Photovoltaic performance of the OSCs. **a** J–V plots of the OSC devices based on DR3TBDTC:PBN-11 and DR3TBDTT:PBN-11 blends under the illumination of AM1.5 G, 100 mWcm$^{-2}$. **b** EQE spectra of the corresponding OSC devices

**Table 2 Photovoltaic parameters of the OSCs based on DR3TBDTC:PBN-11 and DR3TBDTT:PBN-11 blend films**

| Active layers | $V_{OC}$ (V) | $J_{SC}$ (mA cm$^{-2}$) | FF (%) | PCE (%) | $J_{SC}$ (EQE)[a] (mA cm$^{-2}$) |
|---|---|---|---|---|---|
| DR3TBDTC:PBN-11 | 1.11 (1.11 ± 0.01) | 11.18 (11.16 ± 0.23) | 64.6 (64.1 ± 1.4) | 8.01 (7.93 ± 0.06) | 10.63 |
| DR3TBDTT:PBN-11 | 1.15 (1.15 ± 0.01) | 6.21 (6.05 ± 0.26) | 42.9 (41.7 ± 1.4) | 3.06 (2.90 ± 0.12) | 5.92 |

Data in parentheses are the statistical average and error bars of standard deviation calculated from 16 individual devices and data outside of parentheses are the best devices
[a]Integrated values obtained from the EQE spectra

(EQE) spectra of the two devices. The DR3TBDTC:PBN-11 based device shows high photoresponse from 400 to 650 nm with the maximum EQE of 0.61. Based on the integrated EQE spectrum, a calculated $J_{SC}$ of 10.63 mA cm$^{-2}$ was gained for DR3TBDTC: PBN-11 system, which agrees well with the measured $J_{SC}$.

The hole and electron mobilities of the two active layers were estimated by SCLC method based on the J–V plots of the hole-only and electron-only devices. For DR3TBDTC:PBN-11 blend, the $\mu_h$ and electron mobility ($\mu_e$) are $6.97 \times 10^{-4}$ and $2.03 \times 10^{-4}$ cm$^2$ V$^{-1}$ s$^{-1}$, resulting in a $\mu_h/\mu_e$ value of 3.43. The DR3TBDTT: PBN-11 blend shows a $\mu_h$ of $1.33 \times 10^{-4}$ cm$^2$ V$^{-1}$ s$^{-1}$ and a $\mu_e$ of $2.95 \times 10^{-4}$ cm$^2$ V$^{-1}$ s$^{-1}$, corresponding to a $\mu_h/\mu_e$ value of 0.45 (Supplementary Figs. 19–20, and Supplementary Table 8). These results indicate that the hole and electron transport in both active layers is balanced. The photocurrent density ($J_{ph}$) versus effective voltage ($V_{eff}$) plots of the two OSC devices were measured to evaluate the charge generation and collection efficiency[41,42] (Supplementary Fig. 21). As the $V_{eff}$ rises, the $J_{ph}$ increases for both the OSC devices. At high $V_{eff}$, the $J_{ph}$ is saturated for the DR3TBDTC-based OSC, but increases continuously for the DR3TBDTT-based OSC. The $J_{ph,SC}/J_{ph,sat}$ values ($J_{ph,SC}$ is the $J_{ph}$ under short-circuit condition, $J_{ph,sat}$ is the $J_{ph}$ at saturation and the $J_{ph}$ at $V_{eff}$ of 3 V was selected as $J_{ph,sat}$ here) for the DR3TBDTC-based and DR3TBDTT-based devices are 86% and 58%, respectively. The larger $J_{ph,sat}$ and the higher $J_{ph,SC}/J_{ph,sat}$ value suggest that charge generation and collection are more efficient in DR3TBDTC-based device than those in

DR3TBDTT-based device, which agrees well with the excellent photovoltaic performance of DR3TBDTC-based device.

**Active layer morphology.** The molecular ordering of DR3TBDTC and DR3TBDTT in the active layers was investigated by 2D-GIWAXS. The 2D-GIWAXS patterns of the DR3TBDTC: PBN-11 blend annealed at different temperature are shown in Supplementary Fig. 22. The 2D-GIWAXS patterns of the active layers under optimal conditions are shown in Fig. 4a, b, and the corresponding in-plane/out-of-plane one-dimensional (1D) line-cuts are shown in Fig. 4c. The DR3TBDTC:PBN-11 blend shows moderate (100) reflection, weak (200), and (300) reflections of DR3TBDTC in the out-of-plane direction as well as multiple reflections in the $q_{xy}$ direction between 0.9 and 1.5 Å$^{-1}$. The strong (010) reflection in the out-of-plane direction suggests that DR3TBDTC adopts face-on orientation in the active layer. In contrast, DR3TBDTC mainly exhibits edge-on orientation in the neat film and shows negligible crystallization in the as-cast blend film (Supplementary Fig. 10a, d). Considering the face-on orientation of PBN-11 chains in the as-cast blend film, we speculate that the PBN-11 chains have induced DR3TBDTC molecules to adopt face-on orientation in the thermal annealing process of the blend film[43]. The DR3TBDTT:PBN-11 blend exhibits multiple higher order (h00) reflections in the out-of-plane direction and distinct (010) reflection in the in-plane direction of DR3TBDTT. This indicates that DR3TBDTT is

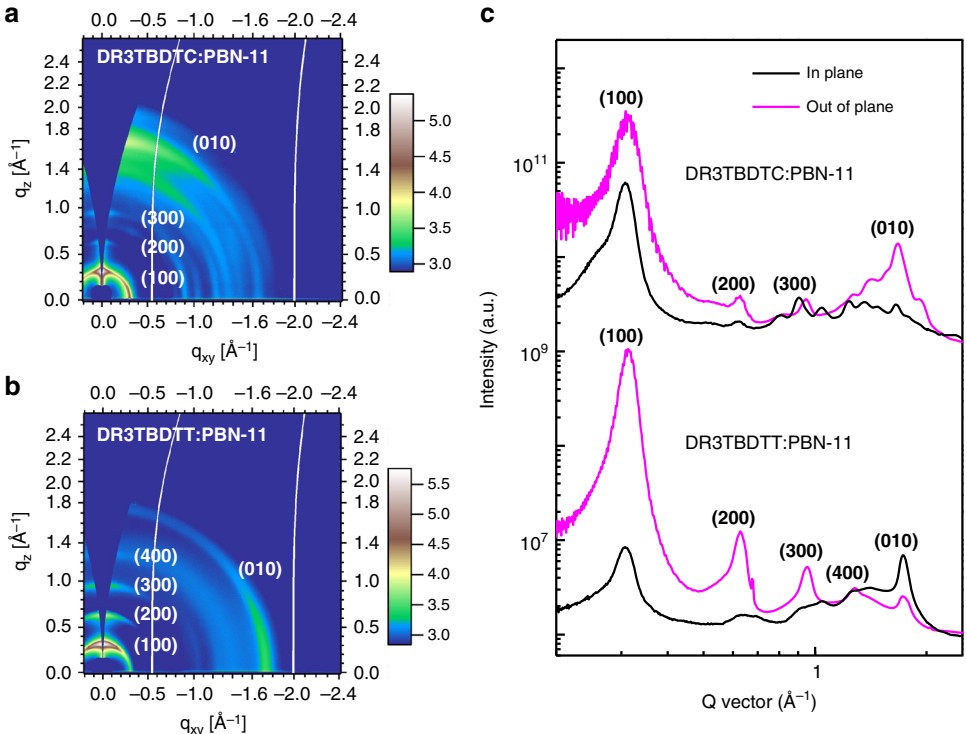

**Fig. 4** GIWAXS data for the blend films. 2D-GIWAXS patterns of **a** DR3TBDTC:PBN-11 and **b** DR3TBDTT:PBN-11 blend films. **c** 1D Linecuts of the corresponding 2D-GIWXAS patterns in the in-plane and out-of-plane directions

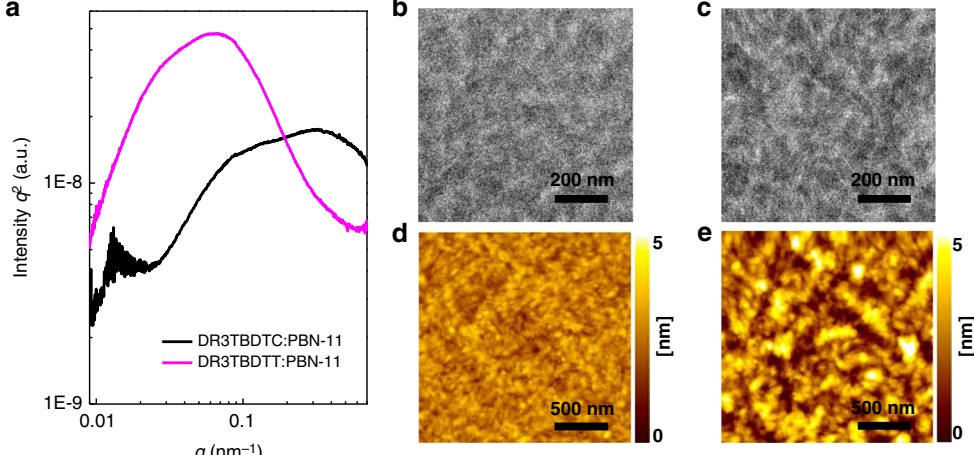

**Fig. 5** Morphology of the blend films. **a** R-SoXS profiles of DR3TBDTC:PBN-11 and DR3TBDTT:PBN-11 blend films. TEM images of **b** DR3TBDTC:PBN-11 and **c** DR3TBDTT:PBN-11 blend films. AFM height images of **d** DR3TBDTC:PBN-11 and **e** DR3TBDTT:PBN-11 blend films

highly crystalline and adopts edge-on orientation in the active layer[27,36]. Compared with the edge-on orientation of DR3TBDTT, the face-on orientation of DR3TBDTC in the $M_D/P_A$ blend can improve the exciton dissociation and the hole transport in the active layer. These results are consistent with the high $J_{ph}$ at saturation and high $\mu_h$ of the DR3TBDTC:PBN-11 blend.

We further investigated the active layer morphology of the two $M_D/P_A$ blends by resonant soft X-ray scattering (R-SoXS), photoluminescence (PL) quenching, transmission electron microscopy (TEM) and atomic force microscopy (AFM). Fig. 5a shows the R-SoXS profiles of the two blends, and the corresponding domain size and domain purity are shown in Supplementary Table 9. The DR3TBDTC:PBN-11 blend shows two domains with

sizes of 30–45 and 8–16 nm. In comparison, the DR3TBDTT: PBN-11 blend exhibits two domains with sizes of 110–130 and 40–55 nm. Considering the high content of small molecule donors in the two blends, we speculate that the relatively large domains are assigned to the donor or/and acceptor-rich phases and the relatively small domains are attributed to the pure small molecule donor phase inside the mixed phases in each blend[44]. In addition, the larger PL quenching efficiency of the donor for DR3TBDTC:PBN-11 blend (89%) than that for DR3TBDTT: PBN-11 blend (80%) also confirms the small-size phase separation in the former case (Supplementary Fig. 23 and Supplementary Table 10). Therefore, we conclude that the suppressed π-π stacking between the molecular backbones of DR3TBDTC leads to small-size phase separation in the $M_D/P_A$

blend. It can greatly increase the donor/acceptor interfacial areas and is very helpful for the exciton dissociation. The relative domain purities of DR3TBDTC:PBN-11 and DR3TBDTT:PBN-11 blend are 0.97 and 1, respectively. It indicates that the suppressed π-π stacking of DR3TBDTC does not obviously decrease the domain purity[45]. Supplementary Fig. 24 shows the TEM images of the DR3TBDTC:PBN-11 blend annealed at different temperature, and Fig. 5b, c show the TEM images of the two active layers under optimal conditions. The DR3TBDTC: PBN-11 blend exhibits interconnected networks of small phase-separated domains, while large-size phase separation appears in the DR3TBDTT:PBN-11 blend. The AFM height images are shown in Fig. 5d, e. The surface of DR3TBDTC:PBN-11 blend film displays lots of small-size aggregates with a low root-mean-square (RMS) roughness of $0.46 \pm 0.08$ nm. In contrast, the surface of DR3TBDTT:PBN-11 blend film shows discontinuous large-size aggregates with a high RMS roughness of $1.12 \pm 0.23$ nm. The interconnected networks in DR3TBDTC:PBN-11 blend benefit the charge transport and collection. The interconnected networks of small phase-separated domains and the face-on orientation can both contribute to the boost in $J_{SC}$ and FF of the DR3TBDTC:PBN-11 based OSC device.

**Thermal stability**. We investigated the thermal stability of DR3TBDTC:PBN-11 based OSCs by thermal annealing the active layer (without deposition of $MoO_3$/Al electrode) at 180 °C for various time. For comparison, we measured the thermal stability of DR3TBDTT:PBN-11 blend and a previously-reported stable $P_D$/$M_A$-type blend of poly[4,8-bis(5-(2-ethylhexyl)thio-phen-2-yl)benzo[1,2-b;4,5-b′]dithiophene-2,6-diyl-alt-(4-(2-ethyl-lhexyl)-3-fluorothieno[3,4-b]thiophene-)-2-carboxylate-2-6-diyl)] (PTB7-Th):(5Z,5′Z)-5,5′-((7,7′-(4,4,9,9-tetrakis(2-ethylhexyl)-4,9-dihydro-s-indaceno[1,2-b:5,6-b′]dithiophene-2,7-diyl)bis(benzo [c][1,2,5]thiadiazole-7,4-diyl))bis(methanylylidene))bis(3-ethyl-2-thioxothiazolidin-4-one) (EH-IDTBR)[46]. The results are shown in Supplementary Fig. 25, and the dependence of PCE on annealing time of the three blends is shown in Fig. 6a. After thermal annealing the active layer at 180 °C for 7 days, the DR3TBDTC:PBN-11 based OSC maintains 89% of the initial PCE, corresponding to the PCE decrease of 11%. The DR3TBDTC:PBN-11 based OSCs with less efficient micro-structures from the suboptimal conditions also exhibit good thermal stability (see Supplementary Fig. 26). The similar excellent thermal stability is also observed for the DR3TBDTT: PBN-11 active layer. In contrast, the PTB7-Th:EH-IDTBR based OSC shows larger PCE decrease in the first 8 h and exhibits the PCE decrease by 45% after 7 days. These results suggest that the DR3TBDTC:PBN-11 and DR3TBDTT:PBN-11 blends

($M_D$/$P_A$-type) are much more thermally stable than the PTB7-Th:EH-IDTBR blend ($P_D$/$M_A$-type).

To study the thermal stability of DR3TBDTC:PBN-11 active layer, we monitored the morphology of the active layer in the thermal annealing process using GIWAXS and TEM. As shown in Supplementary Figs. 27–28, both the GIWAXS patterns and TEM images of the DR3TBDTC:PBN-11 active layer keep nearly unchanged as the thermal annealing is prolonged, indicating excellent morphology stability. This is mainly attributed to the excellent phase stability of the two materials themselves. As shown in Supplementary Fig. 29 and Supplementary Table 11, DR3TBDTC shows the melting temperature ($T_m$) and $T_c$ of above 220 °C, and PBN-11 exhibits no phase transitions in the range from 25 to 300 °C. Moreover, compared with the neat DR3TBDTC, the DR3TBDTC:PBN-11 blend shows negligible depression in $T_m$ and $T_c$, indicating that PBN-11 does not obviously affect the crystallinity of DR3TBDTC in the blend. In comparison, for PTB7-Th:EH-IDTBR blend, EH-IDTBR shows a low $T_c$ of ca. 120 °C, PTB7-Th shows a low glass transition temperature of ca. 140 °C, and the resultant PTB7-Th:EH-IDTBR blend exhibits moderate morphology stability[46,47]. However, when thermal annealed at high temperature of 180 °C, EH-IDTBR may easily diffuse and crystallize in the blend, leading to undesirable morphology and deteriorated photovoltaic performance. Another possible reason for the excellent thermal stability of DR3TBDTC: PBN-11 active layer is the high crystallinity of DR3TBDTC. The crystalline interconnected networks of DR3TBDTC may inhibit the diffusion of the uncrystallized molecules in the active layer during thermal annealing process[48–50].

The light stability of DR3TBDTC:PBN-11 based device was studied by illuminating the devices under 100 mW cm$^{-2}$ AM 1.5 G simulated solar light for different time. Here, we select the PTB7-Th:EH-IDTBR based OSC devices for comparison, which have been reported to show excellent light stability[46]. The results are shown in Supplementary Fig. 30, and the dependence of PCE on illumination time of the two devices is shown in Fig. 6b. The two OSCs show rapid efficiency decline in the first 8 h, which can be assigned to the burn-in loss of the devices. After illumination for 3 days, the DR3TBDTC:PBN-11 device and the PTB7-Th:EH-IDTBR device keep up 74% and 77% of the initial PCE, respectively. This result indicates that the light stability of DR3TBDTC:PBN-11 based OSC is fairly comparable to that of the PTB7-Th:EH-IDTBR based OSC.

## Discussion

In summary, we have demonstrated efficient and thermally stable $M_D$/$P_A$-type OSCs using a small molecular donor with suppressed π-π stacking and a polymer acceptor containing B←N. The

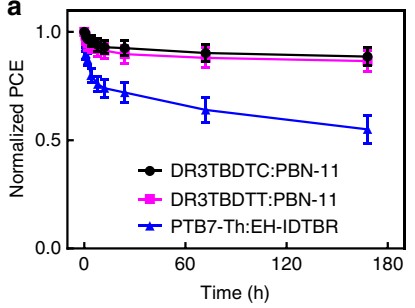
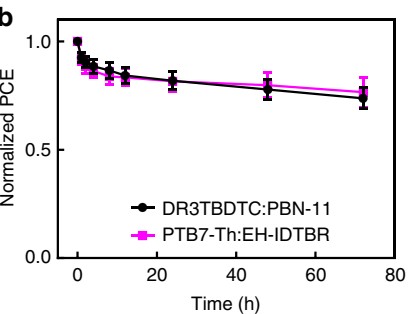

**Fig. 6** Thermal and light stability. **a** The normalized PCE for the OSC devices based on DR3TBDTC:PBN-11, DR3TBDTT:PBN-11, and PTB7-Th:EH-IDTBR blends after annealing the active layers at 180 °C for different time. **b** The normalized PCE for the DR3TBDTC:PBN-11 and PTB7-Th:EH-IDTBR based OSC devices after illumination under 100 mW cm$^{-2}$ AM 1.5 G simulated solar light for different time. All error bars with average values were obtained from six individual devices

$M_D$/$P_A$-type OSC device exhibits small-size phase separation in the active layer and shows the PCE of 8.0%. Moreover, the $M_D$/$P_A$-type OSC exhibits superior thermal stability, i.e. maintaining 89% of its initial PCE after thermal annealing the active layer at 180 °C for 7 days. The demonstration of efficient and stable $M_D$/$P_A$-type OSCs is expected to greatly expand the research field of OSCs.

## Methods

**OSC devices fabrication and measurement.** The OSC devices were fabricated with an inverted structure of ITO/ZnO (40 nm)/active layer/MoO$_3$ (10 nm)/ Al (100 nm). ITO glass substrates were ultrasonicated for 10 min sequentially with detergent, de-ionized water, acetone and iso-propanol, respectively, and then dried at 120 °C for 2 h. The ZnO precursor was synthesized according to the literature[51]. After treated with UV–ozone for 25 min, a thin layer of ZnO was deposited on the pre-cleaned ITO glass substrates through spin coating at 3500 rpm from the precursor solution, and then baked at 200 °C for 60 min in air. All the substrates were transferred to a nitrogen-filled glove box. The optimal donor/acceptor weight ratio for DR3TBDTT:PBN-11 and DR3TBDTC:PBN-11 are 4:1 and 3:1, respectively. The small molecule donor/polymer acceptor blends were dissolved in CB with a fixed polymer concentration of 2.5 mg mL$^{-1}$ in the glove box. The solutions were stirred at 80 °C for 3 h, and then spin-coated onto the ITO/ZnO substrates (preheated at 80 °C) to give the active layer (ca. 90 to 95 nm). After that, the active layers were thermal annealed at 180 °C for 20 min before being transferred into a vacuum chamber. At a pressure of $1 \times 10^{-4}$ Pa, the MoO$_3$ and Al were sequentially deposited on the top of the active layer to complete the devices. The active area of the devices was 8 mm$^2$. For the optimal devices, an aperture with an area of 2 mm$^2$ was also used to measure the performance of the OSCs. A XES-40S2-CE class solar simulator (Japan, SAN-EI Electric Co., Ltd.) was used to provide the AM 1.5 G simulated solar light illumination. The light intensity was calibrated to be 100 mW cm$^{-2}$ using a certified standard monocrystalline silicon (Si) solar cell (SRC-2020, Enli Technology Co., Ltd.). The $J$–$V$ plots of the device were measured with a voltage step of 0.01 V and delay time of 20 ms at 25 °C in a glove box filled with nitrogen (oxygen and water contents are smaller than 0.1 ppm) on a Keithley 2400 source meter. A solar cell spectral response measurement system QE-R3011 (Enli Technology Co., Ltd.) was used to characterize the EQE spectrum under the short-circuit condition. The chopping frequency is 165 Hz.

**Hole- and electron-only devices fabrication and mobility measurements.** The SCLC method was used to characterize the charge mobilities of the $M_D$/$P_A$-blends with the hole-only device of ITO/PEDOT:PSS (40 nm)/active layer/MoO$_3$ (10 nm)/ Al (100 nm) and the electron-only device of ITO/PEIE (10 nm)/active layer/Ca (20 nm)/Al (100 nm), respectively. The $J$–$V$ plots for the hole-only or electron-only devices were measured and fitted to give the mobility by the modified Mott-Gurney equation[52]:

$$J = \frac{9}{8}\varepsilon_r\varepsilon_0\mu\frac{V^2}{L^3}exp\left(0.89\beta\frac{\sqrt{V}}{\sqrt{L}}\right) \quad (1)$$

where $J$ is the current density, $\varepsilon_0$ is permittivity of free space, $\varepsilon_r$ is the relative permittivity of (a $\varepsilon_r$ of 3 was used here), $\mu$ is the charge mobility, $V$ is the potential across the device, $L$ is the thickness of active layer, and $\beta$ is the field-activation factor. The potential $V$ was calculated from the equation: $V = V_{applied} - V_{bi} - V_{series}$, where $V_{applied}$ is the voltage applied to the device, $V_{bi}$ is the relative work function difference between the two electrodes (estimated to be 0 and 0.2 V for hole-holy and electron-only devices), and $V_{series}$ is the series and contact resistance of the device (estimated to be 10–20 Ω from the blank device of ITO/PEDOT:PSS/ MoO$_3$/Al or ITO/PEIE/Ca/Al).

**Thermal and light stability tests.** All the experiments were performed in the nitrogen-filled glove box. For the thermal stability test, the active layers were first thermal annealed at 180 °C for different time, and then MoO$_3$ and Al were deposited onto the active layers to complete the device. For the light stability test, the OSC devices were fabricated using the same condition as the optimal devices, and the XES-40S2-CE class solar simulator with 100 mW cm$^{-2}$ AM 1.5 G simulated solar light illumination was used to irradiate the devices for different time.

## Data availability

The source data underlying Figs. 2–6, Tables 1 and 2, Supplementary Figs. 5, 8–10, 13–18, 23, 25, 26, 30, and Supplementary Table 1 are provided as a Source Data file. The data that support the findings of this study are available from the corresponding author upon reasonable request.

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

## Acknowledgements

We are grateful for the financial support from the National Natural Science Foundation of China (No. 21625403, 51873204, 21875244, 21504066, 21534003), National Key Research and Development Program of China (No. 2018YFE0100600) Funded by MOST and Strategic Priority Research Program of Chinese Academy of Sciences (No. XDB12010200). Z.D. thanks the Youth Innovation Promotion Association of Chinese Academy of Sciences. X-ray data was acquired at beamlines 7.3.3 and 11.0.1.2 at the Advanced Light Source, which is supported by the Director, Office of Science, Office of Basic Energy Sciences, of the U.S. Department of Energy under Contract No. DE-AC02-05CH11231. We thank Chenhui Zhu at beamline 7.3.3, and Cheng Wang at beamline 11.0.1.2 for assistance with data acquisition. We also thank Beijing Synchrotron Radiation Facility (BSRF) for the help on 2D-GIWAXS measurement.

## Author contributions

B.K., X.W., and Y.C. designed and synthesized DR3TBDTC. J.M. and Z.Z. characterized DR3TBDTC. X.L. and C.D. designed and synthesized PBN-11. Z.Z. fabricated OSC devices. Z.Z., Z.D., and J.L. performed TEM and AFM characterization and data analysis. B.L., W.M., J.Z., and Z.D. performed GIWAXS and R-SoXS measurements and data analysis. Z.Z. performed theoretical simulation. Z.D. and J.L. proposed the idea and prepared manuscript. All authors discussed and commented on the paper. J.L., L.W., W.M., and Y.C. supervised the project.

## Additional information

**Competing interests:** The authors declare no competing interests.

