## [Peer Review File · Nature Communications]

Reviewers' comments:

Reviewer #1 (Remarks to the Author):

The manuscript of Wang and co-workers report the synthesis and device fabrication of a novel small molecule donor used in combination with a polymer acceptor, previously reported by the authors. The paper can potentially impact the field of organic photovoltaics for the high thermal stability of the above-mentioned blend at 180 degrees C for 7 days. The manuscript is well constructed and the electrical and morphological characterizations are in well agreement. However the claims reported have to be justified by a significant number of further measurements to be fully supported. As follow a list of points that from my point of view should be addresses in order to consider this manuscript for a high IF journal as Nature Communications. Therefore, at this stage I do not suggest to consider the manuscript for publication.

1) the authors used an inverted device structure based on MoOx/Al as top electrode. It is well known that MoOx severely degrades at high temperature not only in OPV but also in Silicon solar cells. To support the claim of high thermal stability I suggest to run TOF-SIMS and XPS measurements to prove the stability of MoOx.

2) In introduction is written that the morphology of the active layer can be afflicted by thermal stress. The authors do not provide any electrical nor morphological characterization (GIWAXS, SIMS, TEm, ..) of the devices/blends before and after thermal degradation.

3) It is also well known that the solvent additives, i.e. DIO, affect the degradation of the devices. The authors however, chose as references devices, PTB7-Th:PC71BM and PBDB-T:ITIC, in which solvent additives are used. Differently, in their blends no additives have been used. A better comparison is needed to justify their claims.

4) thermal stability is only one part of the picture. At least preliminary results on light stability are necessary.

5) the language of the manuscript is very poor. Nat. Comm. is one of the most read journals in our field and a more appropriate language is of utmost importance.

Reviewer #2 (Remarks to the Author):

In this report, Zhang and coworkers demonstrate an effective solar cell with a small molecule donor and polymer acceptor blend. They synthesized a new small molecule donor similar to DR3TBDTT, except replaced the thiophene sidechain with a carbazole group. With this new molecule, the authors report a higher performance of approximately 8%. Additionally, the authors postulate that this improvement was due to a suppression of crystallinity of the small molecule, introduced via the carbazolyl group. More interestingly, they also demonstrated appropriate thermal stability, thus suggesting an ideal morphology. This work is impactful as there have not been many small molecule donor – polymer acceptor blends for organic solar cells published.

I think that this work demonstrates a very high efficient blend which uses a lesser known small molecule donor and polymer acceptor blend architecture – thus making this work novel and appropriate for publication. However, there are still a variety of issues which need to be addressed before publications. One of the largest issues being: I think the authors focus on the carbazole group suppressing crystallinity is off target and the focus should be on how the carbazole group is able to

change the morphology from edge on to face on. In the comments below, I will address these issues and more in fuller details to help improve the quality and readability of this manuscript.

Comments:

- The authors discuss the impact of steric repulsion a significant amount throughout this text, and one of the key graphics detailing this is part of Figure 1. When comparing the thiophene and the carbazole groups (between the BDT backbone and branched alkyl sidechains), the authors claim that the larger carbazole group “acts as steric hindrance to prevent close stacking of the planar conjugated backbones.” While I certainly agree that the carbazole group is larger than the thiophene group, the authors completely ignore the large and bulky branched side chains which are attached to these groups. Because the authors substitute the side chains with a methyl substituent, I think that their argument is very misleading. I think that a very important question would be: Is there anything special about the carbazolyl group, or can the authors simply use a larger side chain (say C6/C4 instead of C4/C2) on the thiophene unit to achieve the same impact?
- This work focuses a lot on the close packing that can be achieved for these small molecule donors. Because of this, I think that single crystal data would be very helpful, as a crystal structure of BDCA will be more enlightening than the oversimplified DFT calculations.
- To continue with the previous comments, one key claim throughout this text is the carbazole group suppresses crystallinity. As the authors do not have XRD data, the GIWAXS data will be a very important and definitive data point that is needed to support this claim. The DR3TBDTT control small molecule donor has a well defined edge on orientation, which includes the ability to see the (100), (200), and (300) OOP reflections. The new small molecule BDCA still has high crystallinity, especially in the qxy direction between 0.9 and 1.7 Å⁻¹. I think the big difference between the GIWAXS data is the drastic change of morphology (from edge on to face on). I think that the claim the authors should focus on is the carbazolyl group is able to help adopt a face on orientation, which is beneficial for charge transport, thus explaining the boost in Jsc and FF. I think the claim that the crystallinity is suppressed isn't supported by the data.
- For the GIWAXS data, please provide the location (in both reciprocal space and real space) for all the peaks mentioned in the text – examples (100), (200), (300), (010), etc.
- Throughout the text, there is a lack of proper statistics. Without including statistics, the work is inherently less trustworthy, and more importantly, you cannot make any claims about data points being different from one another. Please provide at least standard deviation values for (1) mobility measurements in Table 1, (2) Voc, Jsc, and FF measurements in Table 2, (3) RMS roughness, (4-8) entirety of Table S2, S3, S4, S5, and S6.
- There is a lack of synthetic details, thus making the work difficult to reproduce. Please provide the following: (1) statement about reagents being purchased from commercial vendor and being used without further purification, (2) was THF freshly distilled in your lab, if so how; if not, where was it gotten? (3) what was the solvent system for the column chromatography of compound 2, (4) what was the solvent system for the column chromatography of compound 3, (5) what was the solvent system for the column chromatography of compound 4, (6) what was the solvent system for the recrystallization of compound 5, (7) how was compound 6 prepared – need to provide at least a literature reference or full synthetic details, (8) what was the solvent system for the column chromatography of compound 6, (9) what was the solvent system for the column chromatography of BDCA
- For the Thermal Stability tests, the choice of PTB7-Th:PCBM and PBDB-T:ITIC as comparison are both interesting. While they do have high performance, neither is marked as having high thermal stability – in fact, papers have already shown PTB7-Th:PCBM to be rather unstable (as the authors also show). I think that it would be much more interesting/impactful to either compare to other small molecule donor:polymer acceptor blends (like DR3TBDTT:PBN-11 for example), or blends that have been shown to have better stability (like PTB7-Th:EH-IDTBR for example). I do certainly agree that

BDCA:PBN-11 is a thermally stable system, which is a great quality and reason that this work should be published!

- Naming: I recognize that this is really no proper naming nomenclature in the OPV field as a whole, which allows us to name new molecules/polymers whatever we like, but I think that the name of BDCA can be improved to help increase the readability of this work. First, I am still not completely sure I understand what each letter is referring to. My best guess is BD for benzodithiophene, C for carbazole, and A for acceptor. This completely ignored the three thiophene units and identity of the acceptor. Second, because you are comparing two materials that are very similar, it is weird that they have drastically different names. Based on the naming system used for DR3TBDTT, your molecule (BDCA) would naturally be called DR3TBDTC, for example. Or if you want to completely change the name of the commercially available material, you could rename DR3TBDTT to BDTA (assuming my understanding of 1 was correct). Either way, I think that similar materials should have a similar naming system. I am making this recommendation because I think that it can drastically help the readability of this manuscript. As I was reading through, I had to go back to the structure multiple times to remember which material I was reading about.
- I do not think the authors ever defined CB as chlorobenzene
- Figure 3: (1) axis cut off and can't see number, (2) for the legend, please include the blend (i.e. BDCA:PBN-11) instead of just a single component
- For Figure 4, it would be helpful to include the blend info on the GIWAXS patterns (Fig 4a and 4b) so the reader doesn't have to hunt through the caption to find out which is which. There is plenty of room from qz of 2.0-2.4 to include the "BDCA:PBN-11" label in bolded white text.
- Additionally tables would be helpful to find information, such as (1) domain purity and domain size, (2) PL Quenching data (in fact, only 2 of the 4 quenching efficiencies are written in the text)
- Was the thermal stability tests done only once? It would be ideal if they had a least 3 repeats and that would allow for error bars for the points in Figure 6.
- I would recommend moving the general methods (i.e. lines 262+) to the beginning of the SI. It is a little odd to see them at the end.
- Please include ferrocene on Figure S8
- Why was a 2:1 ratio not tried/shown for the DR3TBDTT:PBN-11 blend?
- Figure S13 caption should be hole-only

Reviewer #3 (Remarks to the Author):

Zhang et al report a small molecular donor / polymeric acceptor composite with interesting solar cell performance and thermal stability. The small molecular donor is a derivative of a previously developed medium bandgap p-type molecule with a slightly more extended sidegroup.

The main finding of the work is an enhanced thermal stability: devices retained about 90 % of their initial performance at 180 ° C for more than 150 hrs. The authors argue that the more amorphous nature of the novel acceptor is responsible for the significantly enhanced stability.

My main concern is that the manuscript is lacking too many details to verify the discussion and conclusion of the manuscript. In general, amorphous systems with high T_g are known to be less sensitive to thermally induced demixing as more crystalline systems. The rationale behind that observation is not fully understood, but is discussed in terms of crystallization enthalpy vs mixing entropy contributions to the chemical potential. Nevertheless, that discussion can not be directly transferred to novel material systems without an in-depth analysis.

The authors are recommended to analyze the origin of the thermal stability in much more detail.

(1) What is the difference in thermal stability between BDCA

:PBN-11 and DR3TBDTT:PBN-11?

(2) How does performance evolve as a function of post annealing?

(3a) What was the time zero performance of the device(s) and how many devices were tested at 180 C?

(3b) Could it be that less efficient microstructures are more stable?

(4) The optimum mixing ratios of 1:4 and 1:3 are slightly atypical for the more amorphous nature of BDCA - is there an explanation?

(5a) Was thermal annealing also applied to the films for GIWAXS analysis (TEM)? - how did the GIWAXS pattern evolve as a function of annealing temperature? Does that correlate to the evolution of device performance?

(5b) The argument of higher crystallinity for BDCA is not that obvious from the presented GIWAXS data. The higher order peaks of DR3TBDTT indicate somewhat larger crystals (by how much?), but looking at the total in-plane and out-of plane crystallinities the effect does not seem to be that strong. It would be important to check that statement with DSC measurements.

(5c) The polymer PBN-11 seems to impact crystallinity of both components and also appears more crystalline for blends with BDCA? It would be important to check that measurement with DSC melting point depression measurements.

In summary, the manuscript by Zhang et al reports an interesting set of data, but is lacking the necessary level of detail to verify the main conclusions. As such, the paper is rejected.

Response to the Reviewers:

Reviewer #1 (Remarks to the Author):

Comments:

The manuscript of Wang and co-workers report the synthesis and device fabrication of a novel small molecule donor used in combination with a polymer acceptor, previously reported by the authors. The paper can potentially impact the field of organic photovoltaics for the high thermal stability of the above-mentioned blend at 180 degrees C for 7 days. The manuscript is well constructed and the electrical and morphological characterizations are in well agreement. However the claims reported have to be justified by a significant number of further measurements to be fully supported. As follow a list of points that from my point of view should be addresses in order to consider this manuscript for a high IF journal as Nature Communications. Therefore, at this stage I do not suggest to consider the manuscript for publication.

Response: The comments and suggestions from Reviewer 1 are very helpful. We accept all of them and we thank Reviewer 1 very much.

(Note. According to the second reviewer's suggestion, the name of the small molecule donor has changed from BDCA to DR3TBDTC).

Questions.

1. The authors used an inverted device structure based on MoO_x/Al as top electrode. It is well known that MoO_x severely degrades at high temperature not only in OPV but also in Silicon solar cells. To support the claim of high thermal stability I suggest to run TOF-SIMS and XPS measurements to prove the stability of MoO_x.

Response: In our original manuscript, we have not clearly described the testing condition of device stability, which make the reviewer misunderstand it. We are sorry for it. In this manuscript, our aim is to investigate the thermal stability of the active layers, so we perform the thermal annealing test of the active layers without deposition of MoO_x/Al electrode. The device data on thermal stability in the manuscript are from the uncompleted devices (the active layers are thermally annealed). To address the reviewer's concerns, we also thermally annealed the complete devices (with MoO_x and Al on the active layer) at 180 °C to investigate the device stability. As shown in Figure R1, the effect of thermal annealing on the completed device is stronger than that on the uncompleted device. This is due to the severe degradation of MoO_x. This result agrees well with the reviewer's comments. Therefore, only the active layer of DR3TBDTC:PBN-11 shows high thermal stability, the MoO_x is not stable enough when annealed at high temperature.

We revise the description on the thermal annealing conditions to avoid misunderstanding. Please see Line 11–14 on Page 18 of the revised manuscript.

Figure R1 Dependence of the normalized PCE on the time of thermal annealing at 180 °C for the uncomplete device (the active layer is thermally annealed) and the complete device (with MoO_x/Al).

2. In introduction is written that the morphology of the active layer can be afflicted by thermal stress. The authors do not provide any electrical nor morphological characterization (GIWAXS, SIMS, TEM, ...) of the devices/blends before and after thermal degradation.

Response: The GIWAXS patterns and TEM images of the DR3TBDTC:PBN-11 blend before and after thermal degradation are provided and discussed in the revised manuscript. Please see Supplementary Fig. 24 and Supplementary Fig. 25 in the Supplementary Information. Please see the discussion on Line 3–7 on Page 15 of the revised manuscript.

The data are also shown in Figure R2 and R3 as below. The GIWAXS diffraction signals become slightly stronger as thermal annealing process is prolonged. The domain size shown in the TEM images slightly increases as the annealing time increases. These results indicate that the morphology of DR3TBDTC:PBN-11 blend is only slightly affected by thermal stress.

Figure R2 GIWAXS patterns of DR3TBDTC:PBN-11 blends annealed at 180 °C for different time: (a) 20 min, (b) 24 h, (c) 72 h, (d) 168 h.

Figure R3 TEM images of DR3TBDTC:PBN-11 blends annealed at 180 °C for different time: (a) 20 min, (b) 24 h, (c) 72 h, (d) 168 h.

3. It is also well known that the solvent additives, i.e. DIO, affect the degradation of the devices. The authors however, chose as references devices, PTB7-Th:PC71BM and PBDB-T:ITIC, in which solvent additives are used. Differently, in their blends no additives have been used. A better comparison is needed to justify their claims.

Response: Following this suggestion, we choose a well-recognized stable PTB7-Th:EH-IDTBR blend for comparison (Baran, D. *et al. Nat. Commun.* 2018, **9**, 2059). In this device, no solvent additive is used. As shown in the Figure R4, the device based on DR3TBDTC:PBN-11 blend shows better thermal stability than that of the device based on PTB7-Th:EH-IDTBR. The reason for the excellent thermal stability of DR3TBDTC:PBN-11 blend is the phase stability of the two materials themselves.

Please see the data in Fig. 6b. Please see the discussion in Line 13–15 on Page 13 and Line 1–13 on Page 14 of the revised manuscript.

Figure R4 The normalized PCE for the OSC devices based on DR3TBDTC:PBN-11 and PTB7-Th:EH-IDTBR after annealing the active layers at 180 °C for different time.

4. thermal stability is only one part of the picture. At least preliminary results on light stability are necessary.

Response: To address this concerns, we compare the light stability of DR3TBDTC:PBN-11 based device and the well-recognized PTB7-Th:EH-IDTBR based device (Baran *et al. Nat. Commun.* 2018, **9**, 2059). The two devices are illuminated under 100 mW cm⁻² AM 1.5G simulated solar light for three days. The results are shown below. DR3TBDTC:PBN-11 device exhibits comparable light stability to that of the PTB7-Th:EH-IDTBR device, indicating excellent light stability of DR3TBDTC:PBN-11 blend.

In the revised manuscript, the data are provided as Figure 6b in the revised manuscript and Supplementary Fig. 27 in the Supplementary Information. The light stability is discussed in Line 21–23 on Page 15 and Line 1–6 on Page 16 of the revised manuscript.

Figure R5 The normalized PCE for the OSCs based on DR3TBDTC:PBN-11 and PTB7-Th:EH-IDRBR systems after illumination under 100 mW cm^{-2} AM 1.5G simulated solar light for different time.

5. *the language of the manuscript is very poor. Nat. Comm. is one of the most read journals in our field and a more appropriate language is of utmost importance.*

Response: Thanks for your suggestion. We have tried our best to polish the language in the revised manuscript.

Reviewer #2 (Remarks to the Author):

Comments:

In this report, Zhang and coworkers demonstrate an effective solar cell with a small molecule donor and polymer acceptor blend. They synthesized a new small molecule donor similar to DR3TBDTT, except replaced the thiophene sidechain with a carbazole group. With this new molecule, the authors report a higher performance of approximately 8%. Additionally, the authors postulate that this improvement was due to a suppression of crystallinity of the small molecule, introduced via the carbazolyl group. More interestingly, they also demonstrated appropriate thermal stability, thus suggesting an ideal morphology. This work is impactful as there have not been many small molecule donor – polymer acceptor blends for organic solar cells published.

I think that this work demonstrates a very high efficient blend which uses a lesser known small molecule donor and polymer acceptor blend architecture – thus making this work novel and appropriate for publication. However, there are still a variety of issues which need to be addressed before publications. One of the largest issues being: I think the authors focus on the carbazole group suppressing crystallinity is off target and the focus should be on how the carbazole group is able to change the morphology from edge on to face on. In the comments below, I will address these issues and more in fuller details to help improve the quality and readability of this manuscript.

Response: The comments and suggestions from Reviewer 2 are also very helpful. We accept all of them and we thank Reviewer 2 very much.

Regarding the effect of carbazole groups on the molecular orientation from edge-on to face-on. 1) We agree that this point is very important. The face-on orientation contributes a lot to the boost of J_{SC} and FF of the devices. 2) We think that the face-on orientation of DR3TBDTC in the blend is induced by the PBN-11. In the as cast film of the blend, while PBN-11 is crystalline and adopts face-on orientation, DR3TBDTC is amorphous. After the thermal annealing for 20 minutes, DR3TBDTC become crystalline and adopts face-on orientation. Therefore, PBN-11 prevents the crystallization of DR3TBDTC in the film-forming process (spin-coating process) and then induces DR3TBDTC to adopt face-on orientation in the crystallization process (thermal annealing process). This inducing effect has been previously reported in literatures (Zhang *et al. Macromolecules* 2016, **49**, 6987. *Polymer* 2018, **138**, 49).

We discuss the edge-on to face-on change in Line 19–23 on Page 10 and Line 1 on Page 11 of the revised manuscript.

(Note. Following the reviewer's suggestion, the name of the small molecule donor has changed from BDCA to DR3TBDTC).

Questions.

1. The authors discuss the impact of steric repulsion a significant amount throughout this text, and one of the key graphics detailing this is part of Figure 1. When comparing the thiophene and the carbazole groups (between the BDT backbone and branched alkyl sidechains), the authors claim that the larger carbazole group “acts as steric hindrance to prevent close stacking of the planar conjugated backbones.” While I certainly agree that the carbazole group is larger than the thiophene group, the authors completely ignore the large and bulky branched side chains which

are attached to these groups. Because the authors substitute the side chains with a methyl substituent, I think that their argument is very misleading. I think that a very important question would be: Is there anything special about the carbazolyl group, or can the authors simply use a larger side chain (say C6/C4 instead of C4/C2) on the thiophene unit to achieve the same impact?

Response: Follow the reviewer's suggestion, we have synthesized an analogous compound of DR3TBDTT, in which C6/C4 instead of C4/C2 is used as the side chain. The compound is named as DR3TBDTT-BO. Its chemical structure is shown below. With DR3TBDTT-BO as donor and PBN-11 as acceptor, the device shows a PCE of 3.25% (Figure R6 and Table R1). This PCE is only slightly higher than that of the DR3TBDTT-based device, but much lower than that of the DR3TBDTC-based device. Thus, we conclude that the carbazolyl groups plays a crucial role in DR3TBDTC.

The data are provided as Supplementary Fig.14 and Supplementary Table 6 in the Supplementary Information.

Figure R6 The chemical structure of DR3TBDTT-BO, and the J - V plots of the OSC devices based on DR3TBDTT-BO:PBN-11, DR3TBDTT:PBN-11, and DR3TBDTC:PBN-11 blends under the illumination of AM1.5G, 100 mW cm^{-2} .

Table R1 Photovoltaic parameters of the OSCs based on DR3TBDTC:PBN-11, DR3TBDTT:PBN-11 and DR3TBDTT-BO:PBN-11 blend films. Data in parentheses are the statistical average and error bars of standard deviation calculated from 16 individual devices and data outside of parentheses are the best devices. The active area is 2 mm^2 .

Active layers	V_{OC} (V)	J_{SC} (mA cm^{-2})	FF (%)	PCE (%)
DR3TBDTC:PBN-11	1.11 (1.11 \pm 0.01)	11.18 (11.16 \pm 0.23)	64.6 (64.1 \pm 1.4)	8.01 (7.93 \pm 0.06)
DR3TBDTT:PBN-11	1.15 (1.15 \pm 0.01)	6.21 (6.05 \pm 0.26)	42.9 (41.7 \pm 1.4)	3.06 (2.90 \pm 0.12)
DR3TBDTT-BO:PBN-11	1.16 (1.15 \pm 0.01)	6.15 (5.96 \pm 0.26)	45.6 (45.3 \pm 1.4)	3.25 (3.10 \pm 0.10)

2. This work focuses a lot on the close packing that can be achieved for these small molecule donors. Because of this, I think that single crystal data would be very helpful, as a crystal structure of BDCA will be more enlightening than the oversimplified DFT calculations.

Response: We agree that single crystal data would be very helpful. After many attempts, we fail to obtain high-quality single crystals of DR3TBDTC. We have searched literatures and cannot find any single crystal data of small molecule donors with the same conjugated backbone as DR3TBDTT. Therefore, we are unable to provide the single crystal data. We hope the reviewer

understand our situation.

3. To continue with the previous comments, one key claim throughout this text is the carbazole group suppresses crystallinity. As the authors do not have XRD data, the GIWAXS data will be a very important and definitive data point that is needed to support this claim. The DR3TBDTT control small molecule donor has a well defined edge on orientation, which includes the ability to see the (100), (200), and (300) OOP reflections. The new small molecule BDCA still has high crystallinity, especially in the q_{xy} direction between 0.9 and 1.7 \AA^{-1} . I think the big difference between the GIWAXS data is the drastic change of morphology (from edge on to face on). I think that the claim the authors should focus on is the carbazolyl group is able to help adopt a face on orientation, which is beneficial for charge transport, thus explaining the boost in J_{sc} and FF. I think the claim that the crystallinity is suppressed isn't supported by the data.

Response: We thank the reviewer very much for pointing out these issues. We agree on this argument. The GIWAXS data suggest that DR3TBDTC still has high crystallinity. The DSC data of the two compounds also support this argument. As shown in Figure R7, DR3TBDTC exhibits higher crystallization temperature ($T_c = 223\text{ }^\circ\text{C}$) and larger crystallization enthalpy change ($\Delta H_c = 52.2\text{ J g}^{-1}$) than those of DR3TBDTT ($T_c = 209\text{ }^\circ\text{C}$, $\Delta H_c = 32.0\text{ J g}^{-1}$), confirming the high crystallinity of DR3TBDTC.

We agree that the face-on orientation of DR3TBDTC in the blend film greatly contribute to the boost in J_{sc} and FF of DR3TBDTC-based device. DR3TBDTC exhibits edge-on orientation in the neat film, shows negligible crystallization in the as-cast blend film, but exhibits face-on orientation in the thermally-annealed blend film. Considering the face-on orientation of PBN-11 chains in the as-cast blend film, we think that the PBN-11 chains have induced DR3TBDTC molecules to adopt face-on orientation in the thermal annealing process of the blend film (Zhang *et al. Macromolecules* 2016, **49**, 6987. *Polymer* 2018, **138**, 49). In addition, DR3TBDTC exhibits weaker π - π stacking than that of DR3TBDTT, as indicated by the weak long-wavelength absorption of DR3TBDTC both in neat film and in blend films (see Figure R8). This leads to smaller phase separation domain size and contributes to the boost of J_{sc} and FF of DR3TBDTC-based device.

We revise the discussion on the reason for the boost of J_{sc} and FF. Please see Line 4–6 on Page 13 of the revised manuscript. The DSC data are also provided in Fig. 2b of the revised manuscript and Supplementary Table 11 in the Supplementary Information.

Figure R7 The DSC thermogram of DR3TBDTC and DR3TBDTT in nitrogen atmosphere with a scan rate of $10\text{ }^\circ\text{C min}^{-1}$.

Figure R8 Absorption spectra of (a) neat DR3TBDTC and DR3TBDTT films and (b) DR3TBDTC:PBN-11 and DR3TBDTT:PBN-11 blend films.

4. For the GIWAXS data, please provide the location (in both reciprocal space and real space) for all the peaks mentioned in the text – examples (100), (200), (300), (010), etc.

Response: We have provided the location of all the peaks. Please see Fig. 4 in the revised manuscript, Supplementary Fig. 7, Supplementary Fig. 19 and Supplementary Fig. 24 in the revised Supplementary Information.

5. Throughout the text, there is a lack of proper statistics. Without including statistics, the work is inherently less trustworthy, and more importantly, you cannot make any claims about data points being different from one another. Please provide at least for (1) mobility measurements in Table 1, (2) Voc, Jsc, and FF measurements in Table 2, (3) RMS roughness, (4-8) entirety of Table S2, S3, S4, S5, and S6.

Response: We have provided standard deviation values in the revised manuscript. Please see Table 1 and Table 2, Line 1–3 on Page 13 in the revised manuscript, and Table S2, S3, S4, S5, S6 and S8 in the revised Supplementary Information.

6. There is a lack of synthetic details, thus making the work difficult to reproduce. Please provide the following: (1) statement about reagents being purchased from commercial vendor and being used without further purification, (2) was THF freshly distilled in your lab, if so how; if not, where was it gotten? (3) what was the solvent system for the column chromatography of compound 2, (4) what was the solvent system for the column chromatography of compound 3, (5) what was the solvent system for the column chromatography of compound 4, (6) what was the solvent system for the recrystallization of compound 5, (7) how was compound 6 prepared – need to provide at least a literature reference or full synthetic details, (8) what was the solvent system for the column chromatography of compound 7, (9) what was the solvent system for the column chromatography of BDCA.

Response: We have provided all these synthetic details on Page 4–9 in the revised Supplementary Information.

(1) Unless noted, all materials were purchased from commercial suppliers and used as received without further purification, (2) THF was freshly distilled in our lab using the lump of sodium metal, (3) the solvent system for the column chromatography of compound 2 was petroleum ether, (4) the solvent system for the column chromatography of compound 3 was 5:1 (v/v)

petroleum ether–dichloromethane, (5) the solvent system for the column chromatography of compound 4 was 6:1 (v/v) petroleum ether–dichloromethane, (6) the solvent system for the recrystallization of compound 5 was dichloromethane/*n*-hexane, (7) compound 6 was prepared according to the reported literature (Chen *et al. Chem. Mater.* 2011, **23**, 4666), (8) the solvent system for the column chromatography of compound 7 was 1:2 (v/v) petroleum ether–chloroform, (9) the solvent system for the column chromatography of BDCA (DR3TBDTC) was 1:4 (v/v) petroleum ether–chloroform.

7. For the Thermal Stability tests, the choice of PTB7-Th:PCBM and PBDB-T:ITIC as comparison are both interesting. While they do have high performance, neither is marked as having high thermal stability – in fact, papers have already shown PTB7-Th:PCBM to be rather unstable (as the authors also show). I think that it would be much more interesting/impactful to either compare to other small molecule donor:polymer acceptor blends (like DR3TBDTT:PBN-11 for example), or blends that have been shown to have better stability (like PTB7-Th:EH-IDTBR for example). I do certainly agree that BDCA:PBN-11 is a thermally stable system, which is a great quality and reason that this work should be published!

Response: Following the reviewer’s suggestion, we choose the well-recognized thermally stable PTB7-Th:EH-IDTBR blend for comparison. As shown in Figure R9, DR3TBDTC:PBN-11 blend shows better thermal stability than that of the PTB7-Th:EH-IDTBR blend. The thermal stability of the different blends is discussed in “Thermal stability” part of results section of the revised manuscript.

Figure R9 The normalized PCE for the OSC devices based on DR3TBDTC:PBN-11, DR3TBDTT:PBN-11 and PTB7-Th:EH-IDTBR blends after annealing the active layers at 180 °C for different time.

8. Naming: I recognize that this is really no proper naming nomenclature in the OPV field as a whole, which allows us to name new molecules/polymers whatever we like, but I think that the name of BDCA can be improved to help increase the readability of this work. First, I am still not completely sure I understand what each letter is referring to. My best guess is if BD for benzodithiophene, C for carbazole, and A for acceptor. This completely ignored the three thiophene units and identity of the acceptor. Second, because you are comparing two materials that are very similar, it is weird that they have drastically different names. Based on the naming system used for DR3TBDTT, your molecule (BDCA) would naturally be called DR3TBDTC, for

example. Or if you want to completely change the name of the commercially available material, you could rename DR3TBDTT to BDTA (assuming my understanding of 1 was correct). Either way, I think that similar materials should have a similar namingsystem. I am making this recommendation because I think that it can drastically help the readability of this manuscript. As I was reading through, I had to go back to the structure multiple times to remember which material I was reading about.

Response: Following the reviewer's suggestion, we name the new molecule as DR3TBDTC in the revised manuscript.

9. I do not think the authors ever defined CB as chlorobenzene.

Response: We thank the reviewer for pointing out this issue. Following the reviewer's suggestion, we defined CB as chlorobenzene in the revised manuscript. Please see Line 12 on Page 6.

9. Figure 3: (1) axis cut off and can't see number; (2) for the legend, please include the blend (i.e. BDCA:PBN-11) instead of just a single component.

Response: We have corrected the axis cut off and included the blend for the legend in Fig. 3 in the revised manuscript.

10. For Figure 4, it would be helpful to include the blend info on the GIWAXS patterns (Fig 4a and 4b) so the reader doesn't have to hunt through the caption to find out which is which. There is plenty of room from q_z of 2.0-2.4 to include the "BDCA:PBN-11" label in bolded white text.

Response: We have included the blend information on the GIWAXS patterns in Fig. 4a and 4b in the revised manuscript.

11. Additionally tables would be helpful to find information, such as (1) domain purity and domain size, (2) PL Quenching data (in fact, only 2 of the 4 quenching efficiencies are written in the text).

Response: The domain purity and domain size for different blends are summarized in Supplementary Table 9, and the PL quenching data are summarized in Supplementary Table 10.

12. Was the thermal stability tests done only once? It would be ideal if they had a least 3 repeats and that would allow for error bars for the points in Figure 6.

Response: The thermal stability tests has been done 6 repeats. The error bars for the points in Fig. 6 were provided in the revised manuscript.

13. I would recommend moving the general methods (i.e. lines 262+) to the beginning of the SI. It is a little odd to see them at the end.

Response: We have moved the general methods to the beginning of the revised Supplementary Information. Please see Page 1–3 in the revised Supplementary Information.

14. Please include ferrocene on Figure S8.

Response: We have included ferrocene in Supplementary Fig. 9 (the updated order) in the revised Supplementary Information.

15. Why was a 2:1 ratio not tried/shown for the DR3TBDTT:PBN-11 blend?

Response: Following the reviewer's suggestion, 2:1 ratio was tried for the DR3TBDTT:PBN-11 blend. Thus, we added the data of the 2:1 ratio in Supplementary Fig. 12 and Supplementary Table 4.

16. Figure S13 caption should be hole-only.

Response: Following the reviewer's suggestion, Supplementary Fig. 16 (the updated order in the revised Supplementary Information) caption has been changed. The space-charge-limited $J-V$ plots for electron-hole devices of DR3TBDTC:PBN-11 and DR3TBDTT:PBN-11 blend films have been individually displayed as Supplementary Fig. 17.

Reviewer #3 (Remarks to the Author):

Comments:

Zhang et al report a small molecular donor / polymeric acceptor composite with interesting solar cell performance and thermal stability. The small molecular donor is a derivative of a previously developed medium bandgap p-type molecule with a slightly more extended sidegroup.

The main finding of the work is an enhanced thermal stability: devices retained about 90 % of their initial performance at 180 ° C for more than 150 hrs. The authors argue that the more amorphous nature of the novel acceptor is responsible for the significantly enhanced stability.

My main concern is that the manuscript is lacking too many details to verify the discussion and conclusion of the manuscript. In general, amorphous systems with high T_g are known to be less sensitive to thermally induced demixing as more crystalline systems. The rational behind that observation is not fully understood, but is discussed in terms of crystallization enthalpy vs mixing entropy contributions to the chemical potential. Nevertheless, that discussion can not be directly transferred to novel material systems without an in-depth analysis.

The authors are recommended to analyze the origin of the thermal stability in much more detail.

Response: The comments and suggestions from Reviewer 3 are very helpful and motivate us to improve the quality of our manuscript. We accept all of them and we thank Reviewer 3 very much.

We have performed more measurement and get more data, then we analyze the origin of thermal stability. We discover that the excellent thermal stability is due to the excellent phase stability of the polymer acceptor and small molecule donor themselves. The small molecule donor, DR3TBDTC, is crystalline in the active layer and its crystallization temperature (T_c) is as high as 223 °C. The crystalline state with the high T_c of DR3TBDTC lead to the excellent thermal stability. This is different from typical thermally stable active layers reported in literatures, in which the materials are amorphous with high glass transition temperature.

We have greatly improved our manuscript. 1) More data in details are provided, including DSC of the materials and the blends, evolution of GIWAXS patterns and TEM images in the thermal annealing process, and light stability. 2) More in-depth analysis is provided, including the origin of excellent thermal stability, the origin of the boost of device efficiency, and the change of molecular orientation in the blend. 3) Clear conclusion is drawn, i.e. the origin of the excellent thermal stability is the crystalline state and the high crystallization temperature of the small molecule donor.

(Note. According to the second reviewer's suggestion, the name of the small molecule donor has changed from BDCA to DR3TBDTC).

Questions.

1. What is the difference in thermal stability between BDCA:PBN-11 and DR3TBDTT:PBN-11?

Response: To address this concerns, we investigate the thermal stability of DR3TBDTT:PBN-11 blend. As shown in Figure R10, the DR3TBDTT:PBN-11 based device shows a PCE retention value of 86% after annealing the active layer at 180 °C for 168 h. The thermal stability of DR3TBDTT:PBN-11 is fairly comparable to that of DR3TBDTC:PBN-11 blend. Please see the data in Supplementary Fig. 22 in the Supplementary Information and the discussion in Line 8–9 on Page 14 in the revised manuscript.

Figure R10 The normalized PCE of the OSC devices based on DR3TBDTC:PBN-11 and DR3TBDTT:PBN-11 blends after annealing the active layers at 180 °C for different time.

2. How does performance evolve as a function of post annealing?

Response: As the annealing process is prolonged, the performance of DR3TBDTC:PBN-11 based OSC devices slightly degrades. As shown in Figure R11 and Table R2, after annealing the active layer at 180 °C for 7 days, the OSC device shows 8%, 2% and 2% decrease in V_{OC} , J_{SC} and FF, respectively.

These data are provided as Supplementary Fig.18 in the Supplementary Information.

Figure R11 $J-V$ plots of the OSCs based on DR3TBDTC:PBN-11 blend after annealing at 180 °C for different time.

Table R2 Photovoltaic performance of the OSCs based on the DR3TBDTC:PBN-11 blend after annealing at 180 °C for different time. The active area is 8 mm².

Time	V_{OC} (V)	J_{SC} (mA cm ⁻²)	FF(%)	PCE (%)
initial	1.15	10.32	62.7	7.44
40 min	1.15	10.28	62.5	7.39
1 h	1.14	10.26	62.4	7.28
2 h	1.13	10.25	62.3	7.19
4 h	1.12	10.24	62.2	7.13
8 h	1.10	10.23	62.2	7.00
12 h	1.09	10.21	62.1	6.91

24 h	1.09	10.20	61.8	6.88
72 h	1.07	10.18	61.7	6.72
168 h	1.06	10.15	61.4	6.61

3. What was the time zero performance of the device(s) and how many devices were tested at 180 °C?

Response: The time zero performance of the devices based on the three blends were listed in Table R2. In our thermal stability test, 6 devices were tested at 180 °C. Follow the reviewers' suggestions, DR3TBDTT:PBN-11 and a new blend of PTB7-Th:EH-IDTBR have been used as comparison.

These data are provided as Supplementary Fig. 22 in the Supplementary Information.

Table R3 The time zero photovoltaic parameters of the OSCs based on DR3TBDTC:PBN-11, DR3TBDTT:PBN-11 and PTB7-Th:EH-IDTBR blend films. Data in parentheses are the statistical average and error bars of standard deviation calculated from 16 individual devices and data outside of parentheses are the best devices. The active area is 8 mm².

Active layers	V_{OC} (V)	J_{SC} (mA cm ⁻²)	FF (%)	PCE (%)
DR3TBDTC:PBN-11	1.15 (1.15 ± 0.01)	10.32 (10.31 ± 0.24)	62.7 (61.9 ± 1.3)	7.44 (7.32 ± 0.06)
DR3TBDTT:PBN-11	1.18 (1.18 ± 0.01)	6.13 (5.99 ± 0.25)	40.1 (39.6 ± 1.3)	2.90 (2.80 ± 0.10)
PTB7-Th:EH-IDTBR	1.03 (1.03 ± 0.01)	15.77 (15.26 ± 0.65)	63.1 (62.2 ± 2.3)	10.24 (9.76 ± 0.35)

4. Could it be that less efficient microstructures are more stable?

Response: To address this concerns, we investigate the thermal stability of the devices based on DR3TBDTC:PBN-11 blends with less efficient microstructures. The less efficient microstructures are obtained by initial thermal annealing at 120 °C and 240 °C (the optimal temperature is 180 °C) during the device fabrication process. As shown in Figure R12, for the device with the initial annealing temperature of 120 °C, the PCE increases dramatically after annealing at 180 °C for 20 minutes and then decreases slowly as the annealing time is prolonged. The devices with initial annealing temperature of 120, 180 and 240 °C maintain 88%, 89% and 92% of the maximal PCE, respectively. The blend annealed at 240 °C (non-optimal) is more stable than the blend with optimal structure (annealed at 180 °C), while the blend annealed at 120 °C (non-optimal) shows similar thermal stability to the blend with optimal structure (annealed at 180 °C).

The data are provided as Supplementary Fig. 23 in the Supplementary Information.

Figure R12 The normalized PCE for the OSC devices based on DR3TBDTC:PBN-11 blends with three initial conditions after annealing the active layers at 180 °C for different time. The DR3TBDTC:PBN-11 blends were first annealed at 120, 180 and 240 °C for 20 minutes before the thermal stability test, respectively.

5. The optimum mixing ratios of 1:4 and 1:3 are slightly atypical for the more amorphous nature of BDCA - is there an explanation?

Response: Optimal mixing ratios of 1:4 and 1:3 are always observed in the blends of amorphous polymer donors and fullerene acceptors. However, in this work, BDCA (DR3TBDTC) is indeed a crystalline compound. This is supported by the diffraction peaks in GIWAXS data and the large crystallization enthalpy change in DSC data of DR3TBDTC (see Figure R13). We are sorry for the unclear description on the crystallinity of DR3TBDTC in the original manuscript, which misleads the reviewer. The optimum mixing ratios of 1:4 and 1:3 are also common for small molecule donor/polymer acceptor type OSCs (Kim *et al.* *Nano Energy*, 2015, **15**, 343; Geng *et al.* *J. Mater. Chem. A*, 2015, **3**, 22325; Zhang *et al.* *Sci. China Chem.* 2018, **61**, 1025). Possibly this is due to the good miscibility of small molecule donors and polymer acceptors, which make the acceptor phase contain certain content of donor (low domain purity).

Figure R13 The DSC thermogram of DR3TBDTC and DR3TBDTT in nitrogen atmosphere with a scan rate of 10 °C min⁻¹.

6. Was thermal annealing also applied to the films for GIWAXS analysis (TEM)? - how did the GIWAXS pattern evolve as a function of annealing temperature? Does that correlate to the

evolution of device performance?

Response: To address this concerns, we perform GIWAXS (at BL1W1A beamline at Beijing Synchrotron Radiation Facility) and TEM characterizations of the DR3TBDTC:PBN-11 blends with different annealing temperature. GIWAXS patterns (see Figure R14) show that the reflection signals become stronger with the increasing annealing temperature, indicating increased crystallinity. TEM images (see Figure R15) shows that the white and dark regions become larger and their contrast becomes obvious as the annealing temperature increases. This result suggests that relatively large and pure domains are formed in the blend with increasing annealing temperature. The change of morphology surely correlates to the change of device performance. The higher crystallinity and purer domains benefit the charge transport and collection, and leads to the improvement in J_{SC} and FF (see Figure R16 and Table R4). However, the much larger domains in the blend annealing at 240 °C limit the exciton dissociation, and thus the J_{SC} decreases.

The GIWAXS and TEM data are provided as Supplementary Fig.19 and Fig. 21 in the Supplementary Information. The evolution of the morphology as the function of annealing temperature is also discussed.

Figure R14 GIWAXS patterns of DR3TBDTC:PBN-11 blends annealing at different temperature for 20 minutes: (a) r.t., (b) 120 °C, (c) 180 °C, (d) 240 °C.

Figure R15 TEM images of DR3TBDTC:PBN-11 blends annealing at different temperature for 20 minutes: (a) r.t., (b) 120 °C, (c) 180 °C, (d) 240 °C.

Figure R16 $J-V$ plots of the OSCs based on DR3TBDTC:PBN-11 blend films annealed at different temperature.

Table R4 Photovoltaic performance of the OSCs based on the DR3TBDTC:PBN-11 blend films annealed at different temperature. Data in parentheses are the statistical average and error bars of standard deviation calculated from 16 individual devices and data outside of parentheses are the best devices. The active area is 8 mm².

Temperature (°C)	V_{oc} (V)	J_{sc} (mA cm ⁻²)	FF (%)	PCE (%)
r.t.	1.22 (1.22 ± 0.01)	1.57 (1.45 ± 0.12)	26.2 (25.6 ± 0.7)	0.50 (0.45 ± 0.03)
80	1.19 (1.19 ± 0.01)	2.98 (2.78 ± 0.19)	34.2 (34.1 ± 1.2)	1.21 (1.13 ± 0.06)
120	1.17 (1.17 ± 0.01)	6.85 (6.64 ± 0.18)	45.4 (44.9 ± 1.2)	3.64 (3.54 ± 0.08)
140	1.16 (1.16 ± 0.01)	8.99 (8.94 ± 0.25)	52.3 (51.6 ± 1.0)	5.46 (5.34 ± 0.11)
160	1.15 (1.15 ± 0.01)	9.91 (9.89 ± 0.23)	59.5 (58.4 ± 1.1)	6.78 (6.63 ± 0.13)
180	1.15 (1.15 ± 0.01)	10.32 (10.31 ± 0.24)	62.7 (61.9 ± 1.3)	7.44 (7.32 ± 0.06)
200	1.15 (1.15 ± 0.01)	10.11 (9.93 ± 0.22)	63.2 (63.0 ± 1.4)	7.35 (7.19 ± 0.10)
240	1.14 (1.14 ± 0.01)	9.89 (9.76 ± 0.19)	63.5 (63.3 ± 1.4)	7.15 (7.04 ± 0.12)

7. The argument of higher crystallinity for BDCA is not that obvious from the presented GIWAXS data. The higher order peaks of DR3TBDTT indicate somewhat larger crystals (by how much?), but looking at the total in-plane and out-of-plane crystallinities the effect does not seem to be that strong. It would be important to check that statement with DSC measurements.

Response: We greatly thank the reviewer for the important comments. To address this concern, we carry out DSC measurements of DR3TBDTC and DR3TBDTT. As shown in Figure R13, DR3TBDTC exhibits higher crystallization temperature ($T_c = 223.3$ °C) and larger crystallization enthalpy change ($\Delta H_c = 52.2$ J g⁻¹) than those of DR3TBDTT ($T_c = 208.6$ °C, $\Delta H_c = 32.0$ J g⁻¹), suggesting the higher crystallinity of DR3TBDTC than DR3TBDTT. Thus, we revise our statement and report that DR3TBDTC also shows high crystallinity. In addition, compared to DR3TBDTT, the weaker π - π stacking of DR3TBDTC leads to small-size phase separation and consequently contributes to enhanced OSC device efficiency.

We discuss the crystallinity of DR3TBDTC in the revised manuscript. Please see Line 17–19 on Page 6 and Line 1–6 on Page 7.

8. The polymer PBN-11 seems to impact crystallinity of both components and also appears more crystalline for blends with BDCA? It would be important to check that measurement with DSC melting point depression measurements.

Response: Follow the reviewer's suggestion, the DSC measurements of the neat DR3TBDTC, DR3TBDTT, PBN-11, the DR3TBDTC:PBN-11 blend (with a ratio of 3:1) and DR3TBDTT:PBN-11 blend (with a ratio of 4:1) blends were carried out. As shown in Figure R17 and Table R5, the DR3TBDTC:PBN-11 blend shows the melting temperature (T_m) and crystallization temperature (T_c) of 253.3 °C and 221.9 °C, which are 2.8 °C and 1.4 °C lower than those of the neat DR3TBDTC, respectively. For the DR3TBDTT:PBN-11 blend, the T_m and T_c are 233.2 and 206.5 °C, which are 1.1 and 2.1 °C lower than those of the neat DR3TBDTT, respectively. The slight melting point depression indicates that the polymer PBN-11 does not

largely impact the crystallinity of DR3TBDTC and DR3TBDTT in the blends. The data are provided and discussed in Supplementary Fig. 25 and Supplementary Table 11 in the Supplementary Information.

We also note that PBN-11 affects the molecular orientation of DR3TBDTC (from edge-on to face-on) in the thermal annealing process of their blend film. Please see the discussion on Line 18-23 on Page 10 of the revised manuscript.

Figure R17 DSC second heating and cooling cycles of neat DR3TBDTC, neat DR3TBDTT, neat PBN-11, DR3TBDTC:PBN-11 blend (3:1, w/w) and DR3TBDTT:PBN-11 blend (4:1, w/w) in N₂ atmosphere with a scan rate of 10 °C min⁻¹.

Table R5 The melting temperature (T_m), crystallization temperature (T_c), melting enthalpy change (ΔH_m) and crystallization enthalpy change (ΔH_c) of the neat DR3TBDTC, DR3TBDTT and PBN-11, and the DR3TBDTC:PBN-11 (3:1) and DR3TBDTT:PBN-11 blends extracted from the 2nd melting and cooling cycles.

Materials	T_m (°C)	ΔH_m (J g ⁻¹)	T_c (°C)	ΔH_c (J g ⁻¹)
DR3TBDTC	256.1	53.5	223.3	52.2
DR3TBDTT	234.3	34.4	208.6	32.0
PBN-11	/	/	/	/
DR3TBDTC:PBN-11	253.3	34.3	221.9	34.2
DR3TBDTT:PBN-11	233.2	26.0	206.5	24.9

REVIEWERS' COMMENTS:

Reviewer #1 (Remarks to the Author):

The authors successfully replied to all my comments. I, therefore, recommend the paper to be published as is.

Reviewer #2 (Remarks to the Author):

In this report, Zhang and coworkers demonstrate an effective solar cell with a small molecule donor and polymer acceptor blend. They synthesized a new small molecule donor similar to DR3TBDTT, except replaced the thiophene sidechain with a carbazole group. With this new molecule, the authors report a higher performance of approximately 8%. Additionally, the authors postulate that this improvement was due to a suppression of crystallinity of the small molecule, introduced via the carbazolyl group. More interestingly, they also demonstrated appropriate thermal stability, thus suggesting an ideal morphology. This work is impactful as there have not been many small molecule donor – polymer acceptor blends for organic solar cells published.

I think that this work demonstrates a very high efficient blend which uses a lesser known small molecule donor and polymer acceptor blend architecture – thus making this work novel and appropriate for publication. Additionally, I believe that the authors made a good effort at addressing the issues raised by reviewers in the original manuscript. There is an appropriate amount of data which supports their claims, however, the language, style, and flow of the manuscript needs improvement. Nature Communications is a highly read journal, and there are many sections that lack clear transitions and include short and choppy sentence structure. For this reason, I recommend minor revisions, with a focus on working on the writing style of the manuscript.

Comments:

- There is a lack of transitions and explanation/discussion on the impact of the results, which hinder the reading and understanding (especially the significance of the data). For example, on page 7, the authors present hole mobilities values, but don't offer any discourse on these numbers. Then, they immediately jump to CV results with no clear transition. For this to be appropriate for Nature Communications, I would recommend the authors go to each section of the manuscript and include the following structure: begin with a clear introductory sentence (i.e. why we are exploring the specific topic), then present the results/data, then include some discourse (i.e. what this means, is this good or bad, how does this compare), then conclude with a transition to what should next be looked at (i.e. the next paragraph).
- Additionally, the sentence structure is very short and choppy, especially in the early parts of the paper.
- There are multiple occurrences of the word "always," which I think should be toned down to "often." These include on line 21, 52
- Also, \bar{M}_w (dispersity) should be used instead of PDI (polydispersity index). The IUPAC (DOI: <https://doi.org/10.1351/PAC-REP-12-03-05>) released a technical report with formal guidelines for polymers nomenclature.
- On line 36, the authors cite high performance OSC at 14%, please update reference to newer publications with Y6, with values now exceeding 16%. (DOI: <https://doi.org/10.1016/j.joule.2019.01.004> ; DOI: [10.1007/s11426-019-9457-5](https://doi.org/10.1007/s11426-019-9457-5))
- There are numerous Supplementary Figures (1, 2, 4, 5, 6, 10, 11, 12, 13, 14, 15, 18, 19, 21, 23, 27, 28, 29, 30) and Tables (1 - 7) which are not referenced in the main text.

Response to the Reviewer:

Reviewer #2 (Remarks to the Author):

Comments:

In this report, Zhang and coworkers demonstrate an effective solar cell with a small molecule donor and polymer acceptor blend. They synthesized a new small molecule donor similar to DR3TBDDT, except replaced the thiophene sidechain with a carbazole group. With this new molecule, the authors report a higher performance of approximately 8%. Additionally, the authors postulate that this improvement was due to a suppression of crystallinity of the small molecule, introduced via the carbazolyl group. More interestingly, they also demonstrated appropriate thermal stability, thus suggesting an ideal morphology. This work is impactful as there have not been many small molecule donor – polymer acceptor blends for organic solar cells published.

I think that this work demonstrates a very high efficient blend which uses a lesser known small molecule donor and polymer acceptor blend architecture – thus making this work novel and appropriate for publication. Additionally, I believe that the authors made a good effort at addressing the issues raised by reviewers in the original manuscript. There is an appropriate amount of data which supports their claims, however, the language, style, and flow of the manuscript needs improvement. Nature Communications is a highly read journal, and there are many sections that lack clear transitions and include short and choppy sentence structure. For this reason, I recommend minor revisions, with a focus on working on the writing style of the manuscript.

Response: We thank the reviewer for the positive comments and valuable suggestions to improve the quality of our manuscript. We have tried our best to improve our writing style and made some changes in revised manuscript using the “track changes” feature. We hope the correction will meet with approval.

Questions:

1. There is a lack of transitions and explanation/discussion on the impact of the results, which hinder the reading and understanding (especially the significance of the data). For example, on page 7, the authors present hole mobilities values, but don't offer any discourse on these numbers. Then, they immediately jump to CV results with no clear transition. For this to be appropriate for Nature Communications, I would recommend the authors go to each section of the manuscript and include the following structure: begin with a clear introductory sentence (i.e. why we are exploring the specific topic), then present the results/data, then include some discourse (i.e. what this means, is this good or bad, how does this compare), then conclude with a transition to what should next be looked at (i.e. the next paragraph).

Response: We thank the reviewer for pointing out this issue. We have added the discourse on the hole mobilities and given the transition to CV results in the revised manuscript. Please see Line 2–5 on Page 8.

2. Additionally, the sentence structure is very short and choppy, especially in the early parts of the

paper.

Response: We thank the reviewer for pointing out this issue. We have revised our manuscript to improve the language using the “track changes” feature.

3. There are multiple occurrences of the word “always,” which I think should be toned down to “often.” These include on line 21, 52.

Response: We sincerely appreciate the valuable comments. As suggested by the reviewer, we have replaced “always” with “often” in the revised manuscript. Please see Line 1 on Page 2 and Line 22 on Page 3.

4. Also, Đ (dispersity) should be used instead of PDI (polydispersity index). The IUPAC (DOI: <https://doi.org/10.1351/PAC-REP-12-03-05>) released a technical report with formal guidelines for polymers nomenclature.

Response: We sincerely appreciate the valuable comments. The “PDI” has been corrected as “Đ” in the revised manuscript. Please see Line 4 on Page 5.

5. On line 36, the authors cite high performance OSC at 14%, please update reference to newer publications with Y6, with values now exceeding 16%. (DOI: <https://doi.org/10.1016/j.joule.2019.01.004> ; DOI:10.1007/s11426-019-9457-5)

Response: We sincerely appreciate the valuable comments. We have updated the references in the revised manuscript. Please see Line 3 on Page 3, and References 4 and 5 in the references part.

6. There are numerous Supplementary Figures (1, 2, 4, 5, 6, 10, 11, 12, 13, 14, 15, 18, 19, 21, 23, 27, 28, 29, 30) and Tables (1 - 7) which are not referenced in the main text.

Response: We thank the reviewer for pointing out this issue. We have referenced these Supplementary Figures and Tables in the revised main text.